# Trimethoprim-Sulfamethoxazole-associated early neutropenia in Mexican adults living with HIV: A cohort study

**Thalia Berenice Jacobo-Vargas**[1,2], **Renata Báez-Saldaña**[3]*, **Luis Pablo Cruz-Hervert**[4,5], **Teresa Imelda Fortoul**[6], **Victor Hugo Ahumada-Topete**[7], **Odalis Rodríguez-Ganén**[8], **Ricardo Stanley Vega-Barrientos**[3]

**1** Pharmacist in Pharmacology Research Unit and Hospital Pharmacy Department, National Institute of Respiratory Diseases, Mexico City, Mexico, **2** Programa de Maestría y Doctorado en Ciencias Médicas, Odontológicas y de la Salud, National Autonomous University of Mexico, Mexico City, Mexico, **3** Hospitalization, National Institute of Respiratory Diseases, Mexico City, Mexico, **4** Professor of Orthodontic Department in the Division of Postgraduate Studies and Research, Faculty of Dentistry, National Autonomous University of Mexico, Mexico City, Mexico, **5** Program on Epidemiological and Emerging Risks, National Autonomous University of Mexico, Mexico City, Mexico, **6** School of Medicine, National Autonomous University of Mexico, Mexico City, Mexico, **7** Head of the Unit of Hospital Epidemiology and Infectious Diseases, National Institute of Respiratory Diseases, Mexico City, Mexico, **8** Head of the Hospital Pharmacy Department, National Institute of Respiratory Diseases, Mexico City, Mexico

* baezrd@unam.mx

## Abstract

## Introduction

Trimethoprim/sulfamethoxazole (TMP/SMX) is the antimicrobial of first choice in the treatment and prophylaxis of *Pneumocystis jirovecii* pneumonia (PCP) in immunocompromised patients, particularly in people living with human immunodeficiency virus (HIV). TMP/SMX use entails different adverse effects, and its association with early neutropenia is minimally documented. This study aimed to identify the risk of early neutropenia associated with TMP/SMX use in adults living with HIV in Mexico.

## Methods

A prospective cohort study was conducted in TMP/SMX-naïve adults living with HIV admitted to a third-level hospital between August 2019 and March 2020. Socio-demographic, clinical, and laboratory data were collected. According to patients' diagnostic, if they required treatment or prophylaxis against PCP, medical staff decided to prescribe TMP/SMX, as it is the first-line treatment. The risk of TMP/SMX induced early neutropenia, as well as associated factors were analyzed through a bivariate model and a multivariate Poisson regression model. The strength of association was measured by incidence rate ratio (IRR) with 95% confidence interval.

## Results

57 patients were enrolled in the study, of whom 40 patients were in the TMP/SMX treatment-group for treatment or prophylaxis of PCP (204.8 person-years of observation, median

**Funding:** This research was supported by the National Council of Science and Technology (CONACyT). TBJ-V is a doctoral student from Programa de Doctorado en Ciencias Médicas, Odontológicas y de la Salud, National Autonomous University of Mexico (UNAM), and received a fellowship from CONACyT (CVU 1196087).

**Competing interests:** The authors have declared that no competing interests exist.

26.5 days) and 17 patients were in the non-treatment group because they did not need the drug for treatment or prophylaxis of PCP (87.0 person-years of observation, median 21 days). The incidence rate of early neutropenia in the TMP/SMX-treatment group versus non-treatment group was 7.81 and 1.15 cases per 100 person-years, respectively. After adjusting for stage 3 of HIV infection and neutrophil count <1,500 cells/mm$^3$ at hospital admission, the current use of TMP/SMX was not associated with an increase in the incidence rate ratio of early neutropenia (adjusted IRR: 3.46; 95% CI: 0.25–47.55; p = 0.352).

## Conclusions

The current use of TMP/SMX in Mexican adults living with HIV was not associated with an increase in the incidence rate ratio of early neutropenia.

## Introduction

Trimethoprim/sulfamethoxazole (TMP/SMX) is the first antimicrobial of choice in the treatment and prophylaxis of *Pneumocystis jirovecii* pneumonia (PCP) in immunocompromised patients, especially in those living with Human Immunodeficiency Virus (HIV). Although the introduction of an antiretroviral therapy (ART) has reduced the morbidity and mortality associated with opportunistic infections, PCP remains one of the most common diagnoses in patients living with HIV [1].

TMP/SMX synergistically blocks bacterial and fungal synthesis of folic acid, a vital cofactor in the production of thymidine and purines [2, 3]. In terms of safety, it is known that people living with HIV have a high frequency of adverse reactions due to TMP/SMX use (20–85%). In addition to different types of cytopenia (anemia, neutropenia, and thrombocytopenia), the appearance of which can range from 6 days to 5 weeks (36), rash (30–55%), elevation of liver enzymes (20%) and of creatinine (1–5%), hyperkalemia, and hyponatremia may occur, usually around day 7 to 14 of treatment [4, 5].

Studies reporting the association of neutropenia with TMP/SMX use are limited. In a cross-sectional study, Fekene *et al.* (2018) reported a significant association between prophylactic use of TMP/SMX and leukopenia (Odds Ratio (OR) 2.34, 95% CI 1.05–5.19, p = 0.036) [6]; on the other hand, Munyazesa *et al.* (2012) did not find a significant association of TMP/SMX with moderate neutropenia (OR 5.69, 95% CI 0.63–51.45; p = 0.122) [7]. While Moh *et al.* (2005) found that grade 3–4 neutropenia was higher in patients with positive serum hepatitis B antigen at baseline (HR 1.58, 95% CI 1.00–2.52; p = 0.05), and compared with patients with a baseline neutrophil count ≥1500 cells/mm$^3$, those with baseline neutrophils al 750–999 cells/mm$^3$ and 1000–1499 cells/mm$^3$ had an HR of grade 3–4 of neutropenia of 3.24 (95% CI 1.94–5.42; p< 0.001) and 2.31 (95% CI 1.55–3.44; p< 0.001), respectively [8]. And in a 6-year longitudinal follow-up study of a cohort of patients living with HIV receiving TMP/SMX prophylaxis, the authors found that the only variable associated with neutropenia was a low baseline CD4 T-cell count [9], which determine the stage of HIV infection. In Mexico, we only found one study in 15 patients living with HIV which found that TMP/SMX dose over 160/800 milligrams respectively, was a risk factor for dermatological adverse reactions to TMP/SMX (OR 12.7, 95% CI 1.59–102.7; p = 0.017) but does not mention whether the patients were new TMP/SMX users [10].

No studies have been conducted in Mexican population, even though neutropenia may be a risk factor for serious bacterial infections. Thus, the objective of this study was to identify the risk of early neutropenia caused by TMP/SMX use in adult patients living with HIV in a third-level hospital in Mexico City.

## Methods

### Study design

A prospective cohort design was used to assess the differences in the risk of early neutropenia between TMP/SMX-treatment group and non-treatment group for prophylaxis or treatment of PCP during the period of current use (while taking the drug), in adults living with HIV hospitalized between August 2019 and March 2020 at the National Institute of Respiratory Diseases "Ismael Cosio Villegas" (INER) in Mexico City. We selected hospitalized patients as our study population because complete blood count (CBC) studies are performed almost daily in the hospital, and we need this information to identify neutropenia events occurring early after initiation of TMP/SMX. To control for the possible bias of being hospitalized, in the multivariate analysis we considered the variable stage 3 of HIV infection at hospital admission as a potential confounder.

### Cohort definition and data collection

Inclusion criteria were hospitalized patient aged 18 years or older, with confirmed diagnosis of HIV, without neutropenia at the start of the study, or with grade 1 or grade 2 neutropenia according to the Common Terminology Criteria for Adverse Events classification version 5.0 [11] and with no TMP/SMX prescription in the 6 months prior to start date (all patients were TMP/SMX-naïve or new users). This washout period was selected to ensure that patients previously exposed to TMP/SMX returned to a naïve state. Patients who had used any chemotherapeutic drug during the past 6 months before the beginning of the study were excluded, as well as those with chronic kidney disease or acute-on-chronic kidney disease, severe aplastic anemia or agranulocytosis and with less than 3 days of TMP/SMX use. Patients who did not have CBC studies done at least 10 days after the start of follow-up were also excluded, because cytopenias are reported to occur in 6 days to 5 weeks [12], and we focused on studying early neutropenia events occurring soon after the start of TMP/SMX, within the first 5 weeks, for treatment or prophylaxis of PCP.

Previous exposure to TMP/SMX or any chemotherapeutic drug during the past 6 months before the beginning of the study was identified by patient interview and by reviewing clinical records because there are not electronic healthcare databases in different health services nor between them.

According to patients' diagnostic, if they required treatment or prophylaxis against PCP, medical staff decided to prescribe TMP/SMX, as it is the first-line treatment. The definition of the TMP/SMX-treatment group were patients with TMP/SMX administered at any dose for at least 3 days, intravenous and/or oral, according to the manual administration records as the source of information at the hospital. Non-treatment group were patients without any TMP/SMX dose administration because they did not have a diagnosis of PCP and therefore did no need TMP/SMX. Patients in the TMP/SMX-treatment group were excluded from entering the non-treatment group at any time.

Consecutive sampling was used to select patients who met the inclusion criteria. Patient information was obtained from daily manual review of prescription and administration records and during the medical visit from Monday to Friday to record drug use, results of laboratory studies, diagnosis, and clinical evolution. Follow up after hospital discharge was done

through monthly telephone calls to identify drug use, and in the case of TMP/SMX-treatment group, the date of drug suspension.

The definition of neutropenia case for patients without baseline neutropenia was neutrophil count $<1,500/mm^3$, and for patients with neutropenia at baseline, either grade 1 (neutrophil count $\geq 1,500/mm^3$) or grade 2 (neutrophil count $<1,500$ to $1,000/mm^3$) was a 20% decrease with respect to the initial neutrophil count. The index date of neutropenia cases was the first date on which neutrophils were found to have a value that met the case definition. CBC was done at the INER laboratory in Beckman Coulter's UniCel® DxH 800 Coulter® Cellular Analysis System with an automated, validated method using the impedance technique. The grades of neutropenia for the identified cases were defined according to the absolute neutrophil count using the Common Terminology Criteria for Adverse Events classification, version 5.0 as follows: grade 2, neutrophils $<1,500$ to $1,000/mm^3$; grade 3, neutrophils $<1,000$ to $500/mm^3$; and grade 4, neutrophils $<500/mm^3$ [11].

Information bias (detection) was ruled out because the comparison between the groups with and without early neutropenia showed no statistically significant difference between the number of days of CBC testing (6 days [4–9 days] vs. 5 days [3–7.5 days], p = 0.0981).

Glomerular filtration rate (GFR) was recorded as a quantitative variable; then a categorical variable was generated depending on whether or not GFR met the condition of reaching $<30$ ml / min at patient's admission to the hospital, because from this rate, dose adjustment is required for some drugs that are eliminated via the kidney, such as TMP/SMX. CD4 T-cell count was recorded as a quantitative variable throughout the follow-up, and a dichotomous variable was generated if the lymphocyte count at hospital admission was $<200$ cells/mm$^3$, since this condition determines the definition of stage 3 of VIH infection, which entails an increased risk of more serious opportunistic infections [13]. Diagnoses as well as prescribed and administered medications were identified throughout hospitalization and until hospital discharge. Drugs given as single doses were not considered because when administered once, it is unlikely that they are involved in blood abnormalities. S1 Appendix describes the rest of the variables that were included in the initial exploratory analysis.

## Statistical analysis

The descriptive analysis was carried out depending on the type of variable and its distribution, evaluated by the Shapiro-Wilk test. The continuous variables were presented as mean ± standard deviation (SD) or median (IQR), as appropriate. The incidence rate of early neutropenia was calculated in number of cases per 100 person-years of follow-up.

According to the distribution of the variables involved, the continuous variables (age, body mass index (BMI), hospital stay, GFR, viral load, CD4 T-cell count, number of days of CBC testing) were compared with the dependent variable (neutropenia) using the t-student test or Mann-Whitney U test, whereas the categorical variables (sex, acute kidney injury, neutrophil count $<1,500$ cells/mm$^3$ at admission, risk group for HIV infection, stage 3 of HIV at admission, sepsis, ART at admission, ganciclovir/valganciclovir use) were compared with the dependent variable (neutropenia) using the $X^2$ test or Fisher's exact test.

The association between TMP/SMX use and early neutropenia was estimated using robust Poisson regression model expressed as IRR with 95% confidence interval, adjusted for potential confounder such as stage 3 of HIV infection at hospital admission [4, 14], and for neutrophil count $<1,500$ cells/mm$^3$ at hospital admission because basal neutropenia has been identified as a risk factor for the development of a more severe degree of neutropenia [8].

Data were analyzed using STATA statistical package version 14 (StataCorp LP, College Station, TX, USA).

## Sample size calculation

With 40 patients in the TMP/SMX-treatment group, 17 patients in the non-treatment group, neutropenia in TMP/SMX-treatment group of 40% and in non-treatment group of 5.9%, the power calculation for the cohort study with OpenEpi was 74% [15].

## Ethical considerations

The study was approved by the Ethics in Researchg Committee and the Research Committee of the Ismael Cosio Villegas National Institute of Respiratory Diseases (registration number E10-19). An informed consent form was signed by each study participant.

## Results

### Socio-demographic and clinical characteristics of the study cohort

57 patients were enrolled in the study, of whom 40 patients were in the TMP/SMX-treatment group for treatment or prophylaxis of PCP and 17 patients were in the non-treatment group because they did not need the drug for treatment or prophylaxis of PCP (Fig 1). Mean age was 38.37 ± 10.28 years. Most participants were men (92.98%), BMI was within the normal category reference range (22.62 ± 3.84). Median (IQR) time of HIV diagnosis was 3.7 months (0–82 months), 70.18% were at the HIV risk group of men who have sex with men (MSM), 75.44% of patients were admitted to the hospital with stage 3 of HIV infection. 17.54% of patients (n = 10) presented sepsis during hospitalization and only one patient had positive serum hepatitis B antigen at baseline. There was statistically significant difference between TMP/SMX-treatment group and non-treatment group in variables such as admission in stage 3 of HIV infection, viral load and CD4 T-cell count at admission, number of days of CBC testing during follow-up, and ART at hospital admission (Table 1).

### Socio-demographic and clinical characteristics between the groups with and without early neutropenia

Of the 57 patients of the study cohort, 17 patients developed early neutropenia during the follow-up. The median (IQR) time of HIV diagnosis was 0.17 months (0–74.3 months) in the early neutropenia group and 4.78 months (0–84.07 months) in the non-neutropenia group (p = 0.523). 94.12% of individuals in the early neutropenia group were admitted with stage 3 of HIV infection vs 67.5% in the non-neutropenia group (p = 0.044). Of the 57 patients, only 24.6% (n = 14), belonging to the group without early neutropenia, were on ART at admission. 29.41% of patients in the early neutropenia group presented sepsis during hospitalization vs 12.5% in the non-neutropenia group (p = 0.145) (Table 2).

### Measures of frequency and association of early neutropenia

The 57 patients of the study cohort were followed for 291.84 person-years (median of 22 days, maximum of 111 days). 17 patients developed early neutropenia (29.82%); of them, in the non-treatment group (87.04 person-years of follow-up, median of 29 days and maximum of 103 days), one case of grade 3 neutropenia was identified. The TMP/SMX-treatment group (204.80 person-years of follow-up, median of 12 days and maximum of 111 days of follow-up) comprised the remaining 16 cases of early neutropenia, of which 12 cases were grade 2, 3 cases were grade 3, and 1 case was grade 4 neutropenia. The incidence rate of early neutropenia in the TMP/SMX-treatment group and non-treatment group was 7.81 and 1.15 cases per 100 person-years, respectively (Table 3).

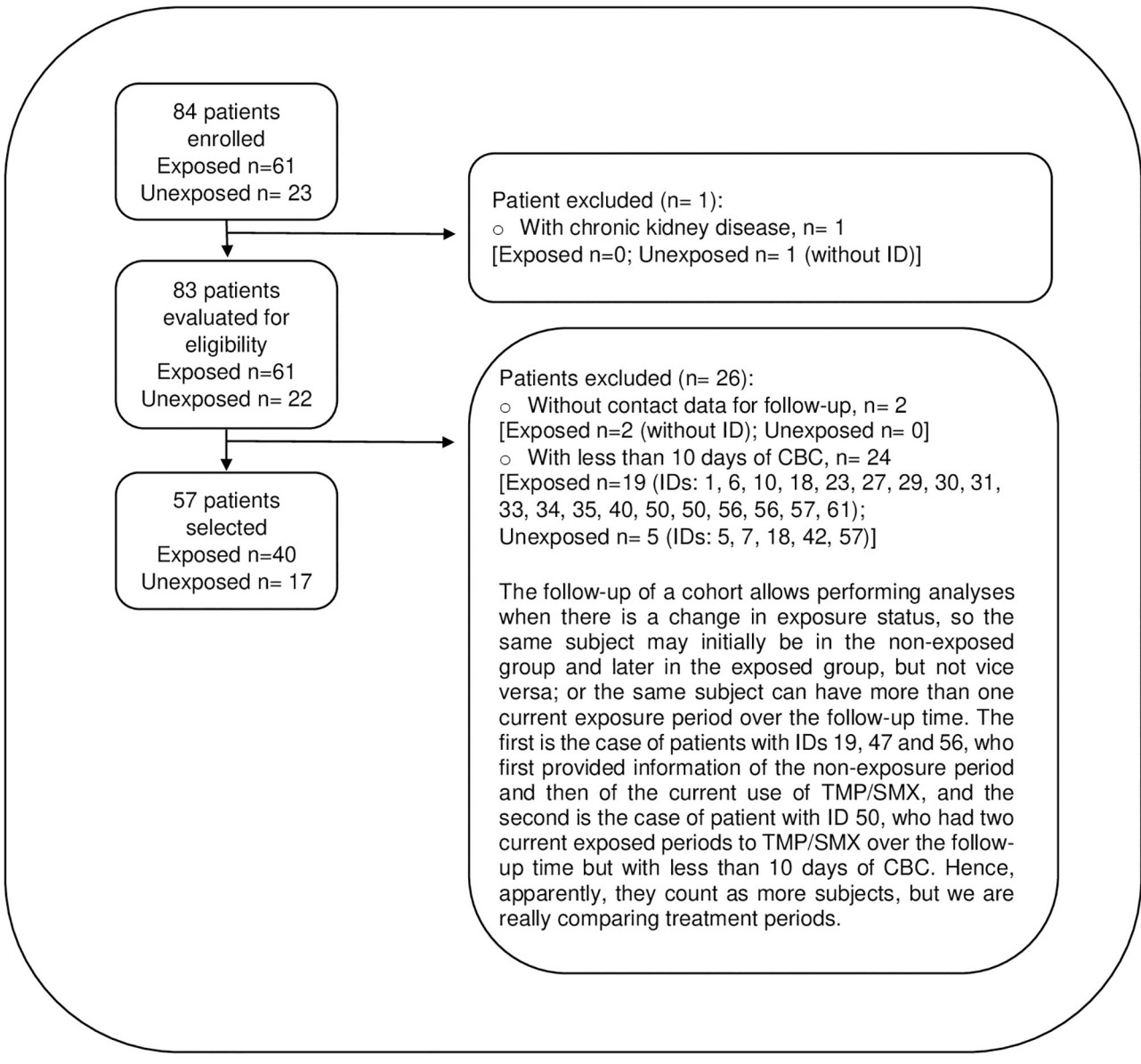

**Fig 1. Flow diagram of patient selection and follow-up.**

After adjusting for stage 3 of HIV infection and neutrophil count <1,500 cells/mm$^3$ at hospital admission, current use of TMP/SMX was not associated with an increase in the incidence rate ratio of early neutropenia (adjusted IRR: 3.46; 95% CI: 0.25–47.55; p = 0.352) (Table 4).

Other clinically relevant variables, such as sepsis during hospitalization as potential confounder, and treatment with ganciclovir/valganciclovir during hospitalization as a drug involved in cytopenias, were not include in the multivariate model because they did not result in a statistically significant difference between the groups with and without early neutropenia in the bivariate model (Table 2). Nor was the variable CD4 T-cell count <200 cells/mm$^3$ at admission included in the multivariate model, because 100% of the patients with early neutropenia (17 cases) had this condition at hospital admission and we have already considered a related variable, stage C3 of HIV infection at hospital admission.

**Table 1. Socio-demographic and clinical characteristics of the total cohort and between TMP/SMX-treatment group and non-treatment group.**

| | Total N = 57 | TMP/SMX-treatment group (n = 40) | Non-treatment group (n = 17) | P value |
|---|---|---|---|---|
| **Age [years], Mean (SD)** | 38.37 (10.28) | 37.25 (9.22) | 41 (12.32) | 0.2701 |
| **Men sex, n (%)** | 53 (92.98) | 37 (92.50) | 16 (94.12) | 1.000 |
| **BMI [kg/m$^2$], Mean (SD)** | 22.62 (3.84) | 21.98 (3.69) | 24.11 (3.88) | 0.0539 |
| **Hospital stay [days], Median (IQR)** | 13 (10,15) | 12 (9.50,16.50) | 14 (11,15) | 0.7725 |
| **GFR at admission [ml/min], Median (IQR)** | 102 (86,121.44) | 105 (83.45,122.93) | 95.35 (88.45,104.20) | 0.2221 |
| **Acute kidney injury at admission, n (%)** | 4 (7.02) | 3 (7.50) | 1 (5.88) | 1.000 |
| **Neutrophil count <1,500 cells/mm$^3$ at admission, n (%)** | 7 (12.28) | 7 (17.50) | 0 | 0.091 |
| **Risk group MSM, n (%)** | 40 (70.18) | 30 (75) | 10 (58.82) | 0.222 |
| **Stage 3 of HIV at admission, n (%)** | 43 (75.44) | 38 (95) | 5 (29.41) | 0.000 |
| **Viral load at admission [copies/ml], Median (IQR)** | 79055 (3869,403558) | 212754.50 (52177,479871) | 47 (1,4117) | 0.0000 |
| **CD4 T-cell count at admission [cells/mm$^3$], Median (IQR)** | 61 (20,188) | 37.50 (19,89.50) | 301 (188,636) | 0.0000 |
| **CD4 T-cell <200 cells/ mm$^3$ at admission, n (%)** | 44 (77.19) | 38 (95.0) | 6 (35.29) | 0.000 |
| **Sepsis, n (%)** | 10 (17.54) | 9 (22.50) | 1 (5.88) | 0.253 |
| **Documented bacterial infection, n (%)** | 25 (43.86) | 16 (40.0) | 9 (52.94) | 0.368 |
| **Parasitic infection, n (%)** | 4 (7.02) | 4 (10.0) | 0 | 0.306 |
| **Number of days of CBC testing, Median (IQR)** | 5 (3,8) | 5.50 (4,9) | 3 (3,5) | 0.0009 |
| **ART at admission, n (%)** | 14 (24.56) | 1 (2.50) | 13 (76.47) | 0.000 |
| **Ganciclovir/Valganciclovir use, n (%)** | 6 (10.5) | 5 (12.5) | 1 (5.88) | 0.657 |

Abbreviations: ART: Antiretroviral therapy; BMI: Body mass index; CBC: Complete blood count; IQR: Inter quartile range; MSM: men who have sex with men; SD: standard deviation; TMP/SMX: trimethoprim/ sulfamethoxazole.

P values were obtained from $X^2$ or Fisher's exact test for categorical variables and t-Student or Mann-Whitney U test for continuous variables.

**Table 2. Socio-demographic and clinical characteristics of the total cohort and between the groups with and without early neutropenia.**

| | Total N = 57 | With neutropenia (n = 17) | Without neutropenia (n = 40) | P value |
|---|---|---|---|---|
| **Age [years], Mean (SD)** | 38.37 (10.28) | 36.35 (10.36) | 39.23 (10.25) | 0.3389 |
| **Men sex, n (%)** | 53 (92.98) | 16 (94.12) | 37 (92.50) | 1.000 |
| **BMI [kg/m$^2$], Mean (SD)** | 22.62 (3.84) | 21.16 (3.62) | 23.23 (3.81) | 0.0620 |
| **Hospital stay [days], Median (IQR)** | 13 (10,15) | 12 (9,15) | 13 (11,16) | 0.6934 |
| **GFR at admission [ml/min], Median (IQR)** | 102 (86,121.44) | 116.0 (81,122.26) | 97.5 (86.75,116.4) | 0.4073 |
| **Acute kidney injury at admission, n (%)** | 4 (7.02) | 2 (11.76) | 2 (5.0) | 0.575 |
| **Neutrophil count <1,500 cells/mm$^3$ at admission, n (%)** | 7 (12.28) | 4 (23.53) | 3 (7.50) | 0.180 |
| **Risk group MSM, n (%)** | 40 (70.18) | 11 (64.71) | 29 (72.50) | 0.547 |
| **Stage 3 of HIV infection at admission, n (%)** | 43 (75.44) | 16 (94.12) | 27 (67.50) | 0.044 |
| **Viral load at admission [copies/ml], Median (IQR)** | 79055 (3869,403558) | 197726 (51256,315974) | 50613 (46,465986.5) | 0.2387 |
| **CD4 T-cell count at admission [cells/mm$^3$], Median (IQR)** | 61 (20,188) | 45 (18,84) | 107.5 (23,284.50) | 0.0292 |
| **CD4 T-cell <200 cells/mm$^3$ at admission, n (%)** | 44 (77.19) | 17 (100.0) | 27 (67.50) | 0.006 |
| **Sepsis, n (%)** | 10 (17.54) | 5 (29.41) | 5 (12.50) | 0.145 |
| **Documented bacterial infection, n (%)** | 25 (43.86) | 5 (29.41) | 20 (50.0) | 0.152 |
| **Parasitic infection, n (%)** | 4 (7.02) | 2 (11.76) | 2 (5.0) | 0.575 |
| **Number of days of CBC testing, Median (IQR)** | 5 (3,8) | 6 (4,9) | 5 (3,7.50) | 0.0981 |
| **ART at admission, n (%)** | 14 (24.56) | 0 | 14 (35.0) | 0.005 |
| **Ganciclovir/Valganciclovir use, n (%)** | 6 (10.5) | 2 (11.76) | 4 (10.0) | 1.000 |

Abbreviations: ART: Antiretroviral therapy; BMI: Body mass index; CBC: Complete blood count; MSM: men who have sex with men.

P values were obtained from $X^2$ or Fisher's exact test for categorical variables and t-Student or Mann-Whitney U test for continuous variables.

**Table 3. Incidence rate and incidence rate ratio of early neutropenia.**

| | | Neutropenia cases | Person-time | Incidence rate (per 100 person-years) | Crude IRR (95% CI) |
|---|---|---|---|---|---|
| Neutropenia 291.84 person-years | TMP/SMX-treatment group | 16 | 204.80 person-years | 7.81 | 6.79 (0.96–48.09) |
| | Non-treatment group | 1 | 87.04 person-years | 1.15 | Ref |

Abbreviations: 95% CI: 95% Confidence Interval; IRR: Incidence rate ratio; TMP/SMX: trimethoprim/ sulfamethoxazole.

A post hoc power analysis was conducted using an online calculator for "A-priori Sample Size for Multiple Regression" [16] to know the minimum required sample size. Considering an anticipated effect size of 0.1, 80% of desired statistical power level, 3 predictors and $\alpha$ = 0.05, the minimum required sample size resulted in 112.

## Discussion

This is the first study to report an estimate of the incidence rate and incidence rate ratio of TMP/SMX-associated early neutropenia in Mexican adults living with HIV. We identified an incidence rate of 7.81 cases of early neutropenia per 100 person-years in the TMP/SMX-treatment group. After adjusting for stage 3 of HIV infection and neutrophil count <1,500 cells/ $mm^3$ at hospital admission we found that the current use of TMP/SMX is not associated with an increase in the incidence rate ratio of early neutropenia (adjusted IRR: 3.46; 95% CI: 0.25–47.55; p = 0.352).

TMP/SMX has been reported to cause neutropenia due to folate deficiency [17]; however, the mechanism has not yet been fully clarified [18]. We found a frequency of neutropenia of 29.82%, similar to that identified by Tamir *et al.* in 2019, who reported a frequency of leukopenia of 24.4% [14]. The difference may be due to variations in the methodology and definitions of cytopenia, as well as different clinical and sociodemographic characteristics of the studied population. In addition, these authors indicate that patients with low CD4 T-cell count, high viral loads, and advanced stages of HIV infection present an increase in the prevalence of leukopenia, which is consistent with our findings, since 100% of patients who developed early neutropenia had CD4 T-cell < 200 cells / $mm^3$ at hospital admission, the median viral load at admission was almost 4 times higher in the group that developed early neutropenia vs the group that did not develop it, and 94% of patients in the early neutropenia group were admitted in stage 3 of HIV infection, unlike the group without early neutropenia, where only 67.5% were admitted in that stage of HIV infection [14]. This indicates that the group of patients who developed early neutropenia had greater degree of immunosuppression at hospital admission.

Although the current use of TMP/SMX in Mexican patients living with HIV was not associated with an increased incidence rate ratio of early neutropenia (adjusted IRR of early neutropenia 3.46; 95% CI 0.25–47.55; p = 0.352), they agree in the direction of the association with

**Table 4. Unadjusted and adjusted measures of association of early neutropenia.**

| | Unadjusted IRR[a] (95% CI) | P value | Adjusted IRR[a] (95% CI) | P value |
|---|---|---|---|---|
| TMP/SMX | 6.80 (0.96–48.09) | 0.055 | 3.46 (0.25–47.55) | 0.352 |
| Neutrophil count <1,500 cells/$mm^3$ at admission | 2.20 (0.99–4.90) | 0.054 | 1.91 (0.87–4.20) | 0.107 |
| Stage 3 of HIV infection at admission | 5.21 (0.74–36.44) | 0.096 | 2.65 (0.26–26.76) | 0.409 |

Abbreviations: 95% CI: 95% Confidence Interval; IRR: Incidence rate ratio; TMP/SMX: trimethoprim/ sulfamethoxazole.
[a]Robust Poisson regression.

those reported by Fekene *et al.* [6], who found in a quantitative cross-sectional study that the prophylactic use of TMP/SMX in adults living with HIV was associated with increased risk of leukopenia (OR 2.34, 95% CI 1.05–5.19, p = 0.036), considering that neutrophils constitute 40–70% of the total leucocyte count [6]. Our results also agree with Munyazesa *et al.*, who did not found association of TMP/SMX and/or dapsone with moderate neutropenia (OR 5.69, 95% CI 0.63–51.45; p = 0.122) [7].

The possible confusion by indication should be considered when finding such associations between antibacterial agents and blood dyscrasias, because antibacterial drugs can be prescribed for the treatment of an infection among whose clinical manifestations are blood dyscrasias [6]. Yet, in the present study, confusion by indication can be ruled out, since PCP pneumonia, for which TMP/SMX was indicated, does not have neutropenia as a clinical manifestation.

Due to the small sample size, we only selected two variables to adjust the IRR of TMP/SMX associated early neutropenia in the multivariate model. Neutrophil count <1,500 cells / mm$^3$ at hospital admission was considered because in the bivariate analysis between the groups with and without early neutropenia, this variable obtained a p = 0.180, also, it modified in >10% the IRR of early neutropenia due to TMP/SMX use (3.02 vs. 4.75), and baseline neutrophil count <1,500 cells / mm$^3$ has been reported to increase the risk of more severe neutropenia (HR 2.31, 95% CI 1.55–3.44; p<0.001) [8]. Additionally, stage 3 of HIV infection at admission was included in the multivariate model, since the advanced stages of HIV infection have been found to be significantly associated with an increase in the prevalence of anemia, leukopenia, and thrombocytopenia [14], and in the bivariate analysis between the groups with and without early neutropenia, stage 3 of HIV infection at admission obtained a p = 0.044. Instead, CD4 T-cell count <200 cells/mm$^3$ was not considered in the adjusted early neutropenia risk model, despite being a confounding factor because 100% of the early neutropenia cases had CD4 T-cell count <200 cells/mm$^3$ at hospital admission, which is consistent with a 6-year longitudinal follow-up study of a cohort of patients living with HIV that used prophylactic TMP/SMX where authors found that the only variable associated with neutropenia was a low baseline CD4 T-cell count [9].

Among the limitations of this study, data about previous medication use could be left truncated, resulting in the incomplete capture of medical history and previous use of medications because there are not healthcare databases in different health services nor between them; this truncation is partially mitigated by comparing information from clinical records with patient interviews. Also, as follow-up for TMP/SMX-treatment group and non-treatment group was carried out until the first neutropenia event occurred or to the time of the last laboratory studies that included CBC, there is potential right truncation because we do not have information of CBC after the discharge of the patients because of the hospital reconversion due to the COVID-19 pandemic. However, the time of follow-up allowed us to identify early neutropenia events. Another limitation of the study is that most patients had a CD4 T-cell count <200 cells/mm$^3$ at hospital admission (77.19%); they came with a high degree of immunosuppression and little control of HIV infection, so the results cannot be generalized to other patients who do not meet these characteristics. Other limitation of the study is the low power obtained for the cohort study (74%) due to the small size of the cohort, as patient enrollment could not continue due to the COVID-19 hospital conversion, suggesting that our true power is even lower for multivariate analysis. However, based on the R2 value of the multivariate model (0.0997), we calculated a small effect size of 0.11 [19]. Although our effect size is small, we can infer that the risk of not identifying a possible effect is very low and additionally, another study has reported no significant effect as mentioned previously. These findings emphasize the need for more studies to identify whether there is a clinically relevance. Furthermore, follow-up

could not be done for a longer time to also identify the risk of neutropenia in the exposure periods of "recent use" and "past use" in addition to "current use". Lastly, the study was developed in a single reference center (unicentric). The early neutropenia association with current TMP/SMX use found in this study may thus be overestimated because of the previous limitations.

Regarding applicability of our estimates to other adult patients living with HIV, even though INER patients included in our cohort study, constitute a different population than that found in other health institutions as INER is a national reference hospital and most patients are admitted with a significant deterioration in their health status, other less advanced stages of HIV infection were also included in addition to stage 3, as well as very varied baseline viral loads, including undetectable patients, both men and women, with a wide age range and from different states of Mexico. All this means that the model can be applied to patients with the variety of characteristics above mentioned.

The results of this study highlight the need to identify groups at high-risk of developing early neutropenia among patients living with HIV with TMP/SMX-treatment. This will allow early pharmacological adjustments and timely initiation of granulocyte colony-stimulating factor therapies to prevent neutropenia or its worsening, which can have fatal consequences. Furthermore, the study contributes to providing evidence that is scarce in the literature on measures of frequency and association of early neutropenia due to TMP/SMX use, even though neutropenia was reported as the most frequent short-term adverse event in a trial of prophylactic TMP/SMX in African adults living with HIV [20]. Finally, our study contributes to the development of pharmacoepidemiology in Mexico.

## Conclusion

The current use of TMP/SMX in Mexican patients living with HIV was not associated with an increased in the incidence rate ratio of early neutropenia. This study highlights important gaps in measures of association of TMP/SMX-associated neutropenia in adults living with HIV. We encourage researchers to continue investigating this area to determine if there is a significant effect that could result in clinical benefit.

## Supporting information

**S1 Appendix. Collected independent variables.**
(DOCX)

**S2 Appendix. STROBE checklist cohort.** STROBE Statement—Checklist of items that should be included in reports of cohort studies [1].
(DOCX)

**S3 Appendix. Study database.**
(DTA)

## Acknowledgments

We would like to thank Dr Gustavo Reyes Teran, Head of the Coordinating Commission of National Institutes of Health and High Specialty Hospitals in Mexico City, Mexico, and to Dr Francisco Javier Flores Murrieta and Dra Miriam del Carmen Carrasco Portugal from National Institute of Respiratory Diseases, Mexico City, Mexico, for all the support given to carry out this study.

Thalia Berenice Jacobo-Vargas is student from the program Doctorado en Ciencias Médicas, Odontológicas y de la Salud at the National Autonomous University of Mexico, Mexico City, Mexico, and is the recipient of scholarship from CONACyT, Mexico (CVU 1196087).

## Author Contributions

**Conceptualization:** Thalia Berenice Jacobo-Vargas, Renata Báez-Saldaña, Luis Pablo Cruz-Hervert, Teresa Imelda Fortoul.

**Data curation:** Thalia Berenice Jacobo-Vargas, Luis Pablo Cruz-Hervert.

**Formal analysis:** Thalia Berenice Jacobo-Vargas, Renata Báez-Saldaña, Luis Pablo Cruz-Hervert, Teresa Imelda Fortoul.

**Investigation:** Thalia Berenice Jacobo-Vargas, Renata Báez-Saldaña, Luis Pablo Cruz-Hervert, Teresa Imelda Fortoul, Victor Hugo Ahumada-Topete.

**Methodology:** Thalia Berenice Jacobo-Vargas, Renata Báez-Saldaña, Luis Pablo Cruz-Hervert, Teresa Imelda Fortoul.

**Project administration:** Thalia Berenice Jacobo-Vargas, Renata Báez-Saldaña, Luis Pablo Cruz-Hervert, Teresa Imelda Fortoul.

**Resources:** Thalia Berenice Jacobo-Vargas, Victor Hugo Ahumada-Topete, Odalis Rodríguez-Ganén, Ricardo Stanley Vega-Barrientos.

**Supervision:** Renata Báez-Saldaña, Luis Pablo Cruz-Hervert, Teresa Imelda Fortoul, Victor Hugo Ahumada-Topete, Odalis Rodríguez-Ganén, Ricardo Stanley Vega-Barrientos.

**Visualization:** Luis Pablo Cruz-Hervert.

**Writing – original draft:** Thalia Berenice Jacobo-Vargas, Renata Báez-Saldaña, Luis Pablo Cruz-Hervert, Teresa Imelda Fortoul, Victor Hugo Ahumada-Topete.

**Writing – review & editing:** Thalia Berenice Jacobo-Vargas, Renata Báez-Saldaña, Luis Pablo Cruz-Hervert, Teresa Imelda Fortoul, Victor Hugo Ahumada-Topete, Odalis Rodríguez-Ganén, Ricardo Stanley Vega-Barrientos.

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
