## [Decision Letter · Decision Letter 0]

6 Feb 2023

PONE-D-22-34622Trimethoprim-Sulfamethoxazole-associated early neutropenia in Mexican adults living with HIV: a cohort study.PLOS ONE

Dear Dr. Jacobo-Vargas,

Thank you for submitting your manuscript to PLOS ONE. After careful consideration, we feel that it has merit but does not fully meet PLOS ONE’s publication criteria as it currently stands. Therefore, we invite you to submit a revised version of the manuscript that addresses the points raised during the review process.

We look forward to receiving your revised manuscript.

Kind regards,

Dickens Otieno Onyango

Academic Editor

PLOS ONE

Journal Requirements:

Reviewers' comments:

Reviewer's Responses to Questions

**Comments to the Author**

1. Is the manuscript technically sound, and do the data support the conclusions?

Reviewer #1: Partly

Reviewer #2: Yes

2. Has the statistical analysis been performed appropriately and rigorously? 

Reviewer #1: Yes

Reviewer #2: I Don't Know

3. Have the authors made all data underlying the findings in their manuscript fully available?

Reviewer #1: Yes

Reviewer #2: Yes

4. Is the manuscript presented in an intelligible fashion and written in standard English?

Reviewer #1: Yes

Reviewer #2: Yes

5. Review Comments to the Author

Reviewer #1: The sample size does not provide adequate power to support the conclusions as well as allow inclusion of all the confounders in the statistical analysis. However, this is captured as a limitation. Additionally, the authors should provide more detailed explanations how the prior results align to their research.

Reviewer #2: Review: TMP/SMX-associated early neutropenia in Mexican adults living with HIV: a cohort study

Overall comments –

• This is an interesting paper, but some of the labeling choices are confusing – specifically, the use of “exposed” and “unexposed” for those who did and did not take TMP/SMX – the label implies previous exposure, whereas what you seem to actually be describing is whether they were or weren’t given the medication after enrolment.

o My suggestion is to be very clear in your methods section that ALL participants entering the study were TMP/SMX-naïve, or as close to possible (>6 months since previous prescription).

o Then describe in your methods how the treatment/non-treatment groups were selected – why were they given this particular antibiotic? Or were they all prescribed it and only 40 agreed to take it? That information doesn’t come through clearly

o You want to make it very clear that the exposure to medication came AFTER enrolment. I’d suggest changing the labeling/language used to ‘treatment-naïve’ for all at enrolment, and then ‘prescribed TMP’ vs ‘not prescribed TMP’ – or even use the terms ‘treatment’ vs ‘non-treatment’ group. But make sure it’s clear that you’re not referring to previous exposure, and eliminate confusion by not using the word ‘exposure’ to mean anything but pre-enrolment use of the medication

Title page –

• Title – capitalize ‘A’ after the colon (:)

• Affiliation #2 – correction needed ‘3’

• Affiliation #3,4 – define “UNAM”

• Affiliation #7 – correction needed ‘4’

• Add corresponding author’s name before email address

Abstract –

• Intro - “…association with early neutropenia has been poorly documented” – consider using ‘minimally documented’ or ‘is not well documented’ – if you’re presenting new information on a new topic, use a phrase that doesn’t imply it’s a well-known concern that just hasn’t been published much

• Methods – remove the word ‘robust’

• Results – ‘stage C3’? Is that ‘clinical stage 3’?

o Final sentence – suggest restating as ‘after adjusting for’ rather than ‘adjusted for’

• Conclusions – remove ‘statistically non-significantly associated’ and just say ‘not associated’.

o Also – your conclusion contradicts your results – your results section “controls for” the variable <1500 cells/mm3, but your conclusion sentence refers to this as a key exposure variable, not something that you’ve controlled for.

• Methods vs Results – The title and methods both call this a cohort study, but your language choice in the results section indicates case and control groups. If this is a cohort study, I suggest rephrasing to indicate “57 patients were enrolled in the study, of whom 40 were previously on TMP/SMX and 17 were TMP/SMX-naïve” (if that’s what you mean).. but it’s not clear in the abstract if your ‘non-exposed’ group was not taking TMP/SMX as their choice (it’s a standard of care?), or whether they were assigned to a no-treatment group. Perhaps clarify in your methods.

Introduction

• Remove ‘AIDS’ and just insert HIV (more politically correct)

• Considering expanding a little more on the known risk factors for neutropenia in new TMP/SMX users – or if not known, add risk factors for other side effects and clearly state that it’s not known for TMP/SMX

Methods

• Study design – remove ‘new user’ before prospective cohort design

o The enrolment of hospitalized patients immediately makes the reader ask ‘why are they hospitalized and is that introducing any bias to the study’ – consider how to refine this section to clarify these points.

• Cohort definition – by ‘start date’ do you mean ‘study enrolment’?

• “early neutropenia events occurring early after the start…” – restate and include “within the first 5 (?) weeks” or something similarly specific

• Sampling: This seems more like consecutive sampling based on daily review of records (not convenience sampling)

• “potential right truncation” – this sentence belongs in your paragraph on study strengths and weaknesses in the discussion section.

• “definition of AIDS stage” – suggest removing ‘AIDS’ and use ‘clinical stage 4’ or similar

• Sample size calculation – you don’t have 1:1 ratio – you have 40:17, which is >2:1

o Suggest revising sample size calculations – or use a website calculator to help you determine your actual study power –

o https://www.openepi.com/Power/PowerCohort.htm - this calculator indicates your actual power is right at 80% (if I understood your parameters correctly) – you can state in the paper your actual power with the sample size you got instead of your original sample size calculations

Results:

• Table 1 – format your table according to journal requirements – check other papers in the journal for examples

• Use sub-heading formatting for the sections within your results section – see examples from published articles in the journal (ie. Bold, bold + Italics, italics + underline – not all the same formatting as the main section titles)

• Sentences at the end of page 14 which talk about which variables were and were not included in the multivariable model belong in your ‘statistical analysis’ section inside the methods.

Discussion

• Don’t use the phrase “statistically non-significantly associated” – just say ‘not associated with’

• You mention you didn’t have the study power you needed because of a small sample size – this contradicts your sample size statements in the methods – make sure they’re consistent – report your actual study power in the methods section (use calculator link provided if helpful) – if the study power was low, then include that statement in your limitations in this section.

• Final sentence – this sentence doesn’t make sense – you don’t need further evidence to generate a larger sample size. Rephrase to make your point more clearly.

• Consider separating a ‘conclusion’ section from the larger discussion section. Make sure this final section comprehensively and concisely summarizes the points you want the reader to remember.

6. PLOS authors have the option to publish the peer review history of their article (what does this mean?). If published, this will include your full peer review and any attached files.

Reviewer #1: No

Reviewer #2: **Yes: **Beth A Tippett Barr

---

## [Author Response · Author response to Decision Letter 0]

22 Mar 2023

March 22, 2023

PlosOne editor,

Thank you for your comments on the manuscript “Trimethoprim-Sulfamethoxazole-associated early neutropenia in Mexican adults living with HIV: A cohort study”. They were very helpful. Below are the answers to each point raised:

Response to Reviewer #1: 

1. The sample size does not provide adequate power to support the conclusions as well as allow inclusion of all the confounders in the statistical analysis. However, this is captured as a limitation. Additionally, the authors should provide more detailed explanations how the prior results align to their research.

We added in the Discussion section: 

Our results also agree with Munyazesa et al., who did not found association of TMP/SMX and/or dapsone with moderate neutropenia (OR 5.69, 95% CI 0.63-51.45; p=0.122) [7].

Also, we have changed the beginning of the previous mentioned paragraph, to read as follows:

Although the current use of TMP/SMX in Mexican patients living with HIV was not associated with an increased incidence rate ratio of early neutropenia (adjusted IRR of early neutropenia 3.46; 95% CI 0.25-47.55; p= 0.352), they agree in the direction of the association with those reported by Fekene et al. [6], who found in a quantitative cross-sectional study that the prophylactic use of TMP/SMX in adults living with HIV was associated with increased risk of leukopenia (OR 2.34, 95% CI 1.05-5.19, p= 0.036), considering that neutrophils constitute 40-70% of the total leucocyte count [6]. Our results also agree with Munyazesa et al., who did not found association of TMP/SMX and/or dapsone with moderate neutropenia (OR 5.69, 95% CI 0.63-51.45; p=0.122) [7]. 

Response to Reviewer #2: 

1. Overall comments:

This is an interesting paper, but some of the labeling choices are confusing – specifically, the use of “exposed” and “unexposed” for those who did and did not take TMP/SMX – the label implies previous exposure, whereas what you seem to actually be describing is whether they were or weren’t given the medication after enrolment.

1.1 My suggestion is to be very clear in your methods section that ALL participants entering the study were TMP/SMX-naïve, or as close to possible (>6 months since previous prescription).

Response to comment 1.1: We added in the Methods section:

All patients were TMP/SMX-naïve.

The paragraph reads as follows:

Inclusion criteria were hospitalized patient aged 18 years or older, with confirmed diagnosis of HIV, without neutropenia at the start of the study, or with grade 1 or grade 2 neutropenia according to the Common Terminology Criteria for Adverse Events classification version 5.0 [11] and with no TMP/SMX prescription in the 6 months prior to start date (all patients were TMP/SMX-naïve or new users). This washout period was selected to ensure that patients previously exposed to TMP/SMX returned to a naïve state.

1.2 Then describe in your methods how the treatment/non-treatment groups were selected – why were they given this particular antibiotic? Or were they all prescribed it and only 40 agreed to take it? That information doesn’t come through clearly.

Response to comment 1.2: We added in the Method section:

Physicians prescribed TMP/SMX according to the patients’ diagnostic for treatment or prophylaxis of PCP, as it is the first line treatment. 

The paragraph reads as follows:

Physicians prescribed TMP/SMX according to the patients’ diagnostic for treatment or prophylaxis of PCP, as it is the first line treatment. The definition of the TMP/SMX-treatment group were patients with TMP/SMX administered at any dose for at least 3 days, intravenous and/or oral, according to the manual administration records as the source of information at the hospital. Non-treatment group were patients without any TMP/SMX dose administration. Patients in the TMP/SMX-treatment group were excluded from entering the non-treatment group at any time.

1.3 You want to make it very clear that the exposure to medication came AFTER enrolment. I’d suggest changing the labeling/language used to ‘treatment-naïve’ for all at enrolment, and then ‘prescribed TMP’ vs ‘not prescribed TMP’ – or even use the terms ‘treatment’ vs ‘non-treatment’ group. But make sure it’s clear that you’re not referring to previous exposure, and eliminate confusion by not using the word ‘exposure’ to mean anything but pre-enrolment use of the medication.

Response to comment 1.2: In the Inclusion criteria we specified that all patients were “TMP/SMX-naïve” or new users, and we changed “exposed” and “not-exposed” group to “TMP/SMX-treatment group” and “non-treatment group” in throughout the document.

The paragraph of cohort definition in the Methods section reads as follows:

Inclusion criteria were hospitalized patient aged 18 years or older, with confirmed diagnosis of HIV, without neutropenia at the start of the study, or with grade 1 or grade 2 neutropenia according to the Common Terminology Criteria for Adverse Events classification version 5.0 [11] and with no TMP/SMX prescription in the 6 months prior to start date (all patients were TMP/SMX-naïve or new users).

2. Title page:Title – capitalize ‘A’ after the colon (:)

Response to comment 2.1: We added a capitalized “A” after the colon (:)

The Title reads as follows:

Trimethoprim-Sulfamethoxazole-associated early neutropenia in Mexican adults living with HIV: A cohort study.

2.2. Affiliation #2 – correction needed ‘3’

Response to comment 2.2: We corrected the affiliation #2.

The affiliation #2 reads as follows:

Affiliation 2: Hospitalization Service, National Institute of Respiratory Diseases, Mexico City, Mexico.

2.3. Affiliation #3,4 – define “UNAM”

Response to comment 2.3: We defined “UNAM”: National Autonomous University of Mexico.

The affiliations #3 and 4 read as follows:

Affiliation 3: Head of the Division of Postgraduate Studies and Research, School of Dentistry, National Autonomous University of Mexico, Mexico City, Mexico.

Affiliation 4: Coordinator of the Master and PhD Program in Medical, Odontological and Health Sciences, National Autonomous University of Mexico, Mexico City, Mexico.

2.4. Affiliation #7 – correction needed ‘4’

Response to comment 2.4: We corrected the affiliation #7.

The affiliation #7 reads as follows:

Affiliation 7: Hospitalization Service, National Institute of Respiratory Diseases, Mexico City, Mexico.

2.5. Add corresponding author’s name before email address.

Response to comment 2.5: We added the author’s name before email address as follows:

Corresponding author: Renata Báez Saldaña

E-mail: baezrd@unam.mx (RBS)

3. Abstract

3.1. Intro - “…association with early neutropenia has been poorly documented” – consider using ‘minimally documented’ or ‘is not well documented’ – if you’re presenting new information on a new topic, use a phrase that doesn’t imply it’s a well-known concern that just hasn’t been published much.

Response to comment 3.1: We considered using ‘minimally documented’.

The sentence reads as follows:

TMP/SMX use entails different adverse effects, and its association with early neutropenia is minimally documented.

3.2 Methods – remove the word ‘robust’.

Response to comment 3.2: The word ‘robust’ was removed.

The sentence reads as follows:

The risk of TMP/SMX induced early neutropenia, as well as associated factors were analyzed through a bivariate model and a multivariate Poisson regression model.

3.3 Results – ‘stage C3’? Is that ‘clinical stage 3’?

Response to comment 3.3: We corrected to “stage 3” without “C”.

The sentence reads as follows:

After adjusting for stage 3 of HIV infection and neutrophil count <1,500 cells/mm3 at hospital admission…

3.3.1 Final sentence – suggest restating as ‘after adjusting for’ rather than ‘adjusted for’.

Response to comment 3.3.1: We restated the sentence as follows:

After adjusting for stage 3 of HIV infection and neutrophil count <1,500 cells/mm3 at hospital admission, the current use of TMP/SMX was not associated with an increase in the incidence rate ratio of early neutropenia (adjusted IRR: 3.46; 95% CI: 0.25-47.55; p= 0.352).

3.4 Conclusions – remove ‘statistically non-significantly associated’ and just say ‘not associated’. 

Response to comment 3.4: We removed ‘statistically non-significantly associated’ and we just said ‘not associated’.

The paragraph reads as follows:

Current use of TMP/SMX in Mexican adults living with HIV was not associated with an increase in the incidence rate ratio of early neutropenia. 

3.4.1 Also – your conclusion contradicts your results – your results section “controls for” the variable <1500 cells/mm3, but your conclusion sentence refers to this as a key exposure variable, not something that you’ve controlled for.

Response to comment 3.4.1: We corrected the paragraph by deleting: “The results suggest that presenting neutrophil count <1,500 cells/mm3 at hospital admission may increase the incidence rate ratio of a more severe degree of neutropenia due to TMP/SMX use”. Because as you said, we have already controlled for that variable.

The Conclusion paragraph in the Abstract reads as follows:

Current use of TMP/SMX in Mexican adults living with HIV was not associated with an increase in the incidence rate ratio of early neutropenia. 

3.5 Methods vs Results – The title and methods both call this a cohort study, but your language choice in the results section indicates case and control groups. If this is a cohort study, I suggest rephrasing to indicate “57 patients were enrolled in the study, of whom 40 were previously on TMP/SMX and 17 were TMP/SMX-naïve” (if that’s what you mean).. but it’s not clear in the abstract if your ‘non-exposed’ group was not taking TMP/SMX as their choice (it’s a standard of care?), or whether they were assigned to a no-treatment group. Perhaps clarify in your methods.

Response to comment 3.5 in the Methods section: We clarified in the method section of the abstract that patients were “TMP/SMX-naïve”. In addition, we added information about how patients were assigned to the TMP/SMX-treatment group or non-treatment group (according to patients’ diagnostic) and that TMP/SMX is the first-line treatment for treatment or prophylaxis against Pneumocystis jirovecii pneumonia (PCP).

The Methods paragraph in the Abstract reads as follows:

A prospective cohort study was conducted in TMP/SMX-naïve adults living with HIV admitted to a third-level hospital between August 2019 and March 2020. Socio-demographic, clinical, and laboratory data were collected. According to patients’ diagnostic, if they required treatment or prophylaxis against PCP, medical staff decided to prescribe TMP/SMX, as it is the first-line treatment.

Response to comment 3.5 in the Results section: We clarified in the results section of the abstract that 57 patients were enrolled in the study, of whom 40 patients were in the TMP/SMX-treatment group for treatment or prophylaxis of PCP and 17 patients were in the non-treatment group because they did not need the drug for treatment or prophylaxis of PCP.

The Results paragraph in the Abstract reads as follows:

57 patients were enrolled in the study, of whom 40 patients were in the TMP/SMX treatment-group for treatment or prophylaxis of PCP (204.8 person-years of observation, median 26.5 days) and 17 patients were in the non-treatment group because they did not need the drug for treatment or prophylaxis of PCP (87.0 person-years of observation, median 21 days). The incidence rate of early neutropenia in the TMP/SMX-treatment group versus non-treatment group was 7.81 and 1.15 cases per 100 person-years, respectively.

4. Introduction

4.1. Remove ‘AIDS’ and just insert HIV (more politically correct)

Response to comment 4.1: We removed ‘AIDS’ and inserted HIV.

The first paragraph of the Introduction section reads as follows:

Trimethoprim/sulfamethoxazole (TMP/SMX) is the first antimicrobial of choice in the treatment and prophylaxis of Pneumocystis jirovecii pneumonia (PCP) in immunocompromised patients, especially in those living with Human Immunodeficiency Virus (HIV). Although the introduction of an antiretroviral therapy (ART) has reduced the morbidity and mortality associated with opportunistic infections, PCP remains one of the most common diagnoses in patients living with HIV [1].

4.2. Considering expanding a little more on the known risk factors for neutropenia in new TMP/SMX users – or if not known, add risk factors for other side effects and clearly state that it’s not known for TMP/SMX.

Response to comment 4.2: We expanded the Introduction section.

We added in the third paragraph of the Introduction section: 

While Moh et al. (2005) found that grade 3-4 neutropenia was higher in patients with positive serum hepatitis B antigen at baseline (HR 1.58, 95% CI 1.00-2.52; p= 0.05) and compared with patients with a baseline neutrophil count ≥1500 cells/mm3, those with baseline neutrophils al 750-999 cells/mm3 and 1000-1499 cells/mm3 had an HR of grade 3-4 of neutropenia of 3.24 (95% CI 1.94-5.42; p< 0.001) and 2.31 (95% CI 1.55-3.44; p< 0.001), respectively [8]. And in a 6-year longitudinal follow-up study of a cohort of patients living with HIV receiving TMP/SMX prophylaxis, the authors found that the only variable associated with neutropenia was a low baseline CD4 T-cell count [9], which determine the stage of HIV infection. In Mexico, we only found one study in 15 patients living with HIV which found that TMP/SMX dose over 160/800 milligrams respectively, was a risk factor for dermatological adverse reactions to TMP/SMX (OR 12.7, 95% CI 1.59-102.7; p= 0.017) but does not mention whether the patients were new TMP/SMX users [10].

5. Methods

5.1. Study design – remove ‘new user’ before prospective cohort design.

Response to comment 5.1: We removed ‘new user’ before prospective cohort design.

The first sentence of study design in the Methods section reads as follows:

A prospective cohort design was used to assess the differences in the risk of early neutropenia between TMP/SMX-treatment group and non-treatment group for prophylaxis or treatment of PCP during the period of current use (while taking the drug), in adults living with HIV hospitalized between August 2019 and March 2020 in Mexico City at the National Institute of Respiratory Diseases “Ismael Cosio Villegas” (INER).

5.1.1. The enrolment of hospitalized patients immediately makes the reader ask ‘why are they hospitalized and is that introducing any bias to the study’ – consider how to refine this section to clarify these points. 

Response to comment 5.1.1: We clarified why we chose hospitalized patients as our study population.

We added in the Study design in the Methods section:

We selected hospitalized patients as our study population because complete blood count (CBC) studies are performed almost daily in the hospital, and we need this information to identify neutropenia events occurring early after initiation of TMP/SMX. To control for the possible bias of being hospitalized, in the multivariate analysis we considered the variable stage 3 of HIV infection at hospital admission as a potential confounder.

5.2. Cohort definition – by ‘start date’ do you mean ‘study enrolment’? 

Response to comment 5.2: Yes.

5.3. “early neutropenia events occurring early after the start…” – restate and include “within the first 5 (?) weeks” or something similarly specific.

Response to comment 5.3: We restated the sentence and included “within the first 5 weeks”.

The sentence reads as follows:

Patients who did not have CBC studies done at least 10 days after the start of follow-up were also excluded, because cytopenias are reported to occur in 6 days to 5 weeks [12], and we focused on studying early neutropenia events occurring soon after the start of TMP/SMX, within the first 5 weeks, for treatment or prophylaxis of PCP.

5.4. Sampling: This seems more like consecutive sampling based on daily review of records (not convenience sampling).

Response to comment 5.4: Yes, we corrected it to consecutive sampling.

The sentence reads as follows:

Consecutive sampling was used to select patients who met the inclusion criteria.

5.5. “potential right truncation” – this sentence belongs in your paragraph on study strengths and weaknesses in the discussion section.

Response to comment 5.5: We moved the sentence to the discussion section.

We added in the sixth paragraph of the Discussion section: 

Also, as follow-up for TMP/SMX-treatment group and non-treatment group was carried out until the first neutropenia event occurred or to the time of the last laboratory studies that included CBC, there is potential right truncation because we do not have information of CBC after the discharge of the patients because of the hospital reconversion due to the COVID-19 pandemic. However, the time of follow-up allowed us to identify early neutropenia events.

5.6. “definition of AIDS stage” – suggest removing ‘AIDS’ and use ‘clinical stage 4’ or similar

Response to comment 5.6: We removed ‘AIDS’ and used ‘stage 3 of VIH infection’.

The sentence reads as follows:

CD4 T-cell count was recorded as a quantitative variable throughout the follow-up, and a dichotomous variable was generated if the lymphocyte count at hospital admission was <200 cells/mm3, since this condition determines the definition of stage 3 of VIH infection, which entails an increased risk of more serious opportunistic infections [13].

5.7. Sample size calculation – you don’t have 1:1 ratio – you have 40:17, which is >2:1.

5.7.1. Suggest revising sample size calculations – or use a website calculator to help you determine your actual study power –

5.7.2. https://www.openepi.com/Power/PowerCohort.htm - this calculator indicates your actual power is right at 80% (if I understood your parameters correctly) – you can state in the paper your actual power with the sample size you got instead of your original sample size calculations.

Response to comments 5.7.1 and 5.7.2: We calculated the power of the cohort study with the calculator you mentioned, and the power was 74%. We stated in the paper our actual power with the sample size we got instead of our original sample size calculation.

The sample size calculation paragraph reads as follows:

With 40 patients in the TMP/SMX-treatment group, 17 patients in the non-treatment group, neutropenia in TMP/SMX-treatment group of 40% and in non-treatment group of 5.9%, the power calculation for the cohort study with OpenEpi was 74% [15].

6. Results

6.1. Table 1 – format your table according to journal requirements – check other papers in the journal for examples.

Response to comment 6.1: Done. We checked other papers to format our tables according to journal requirements.

6.2. Use sub-heading formatting for the sections within your results section – see examples from published articles in the journal (ie. Bold, bold + Italics, italics + underline – not all the same formatting as the main section titles).

Response to comment 6.2: As indicated in the Manuscript Body Formatting Guidelines: level 1 headings are bold type and 18pt font and level 2 headings are bold type and 16pt font.

6.3. Sentences at the end of page 14 which talk about which variables were and were not included in the multivariable model belong in your ‘statistical analysis’ section inside the methods.

Response to comment 6.3: In the statistical analysis section we added the continuous and categorical variables included.

The second paragraph of the statistical analysis section reads as follows: 

According to the distribution of the variables involved, the continuous variables (age, body mass index (BMI), hospital stay, GFR, viral load, CD4 T-cell count, number of days of CBC testing) were compared with the dependent variable (neutropenia) using the t-student test or Mann-Whitney U test, whereas the categorical variables (sex, acute kidney injury, neutrophil count <1,500 cells/mm3 at admission, risk group for HIV infection, stage 3 of HIV at admission, sepsis, ART at admission, ganciclovir/valganciclovir use) were compared with the dependent variable (neutropenia) using the X2 test or Fisher's exact test.

7. Discussion

7.1. Don’t use the phrase “statistically non-significantly associated” – just say ‘not associated with’.

Response to comment 7.1: We removed ‘statistically non-significantly associated’ and we just said ‘not associated’.

The sentence reads as follows:

After adjusting for stage 3 of HIV infection and neutrophil count <1,500 cells/mm3 at hospital admission we found that the current use of TMP/SMX is not associated with an increase in the incidence rate ratio of early neutropenia.

7.2. You mention you didn’t have the study power you needed because of a small sample size – this contradicts your sample size statements in the methods – make sure they’re consistent – report your actual study power in the methods section (use calculator link provided if helpful) – if the study power was low, then include that statement in your limitations in this section.

Response to comment 7.2: We added the power of the cohort study in the methods section. Also, we mentioned it as a limitation of the study in the discussion section.

We added in the sixth paragraph of the Discussion section: 

Other limitation of the study is the low power obtained for the cohort study (74%) due to the small size of the cohort, as patient enrollment could not continue due to the COVID-19 hospital conversion, suggesting that our true power is even lower for multivariate analysis. However, based on the R2 value of the multivariate model (0.0997), we calculated a small effect size of 0.11. Although our effect size is small, we can infer that the risk of not identifying a possible effect is very low and additionally, another study has reported no significant effect as mentioned previously. These findings emphasize the need for more studies to identify whether there is a clinically relevance.

7.3. Final sentence – this sentence doesn’t make sense – you don’t need further evidence to generate a larger sample size. Rephrase to make your point more clearly.

Response to comment 7.3: We have moved the last paragraph from the Discussion section to the Conclusion section and rephrased the sentence as shown in point 7.4.

7.4. Consider separating a ‘conclusion’ section from the larger discussion section. Make sure this final section comprehensively and concisely summarizes the points you want the reader to remember.

Response to comment 7.4: We added a Conclusion section.

The Conclusion paragraph reads as follows:

The current use of TMP/SMX in Mexican patients living with HIV was not associated with an increase in the incidence rate ratio of early neutropenia. This study highlights important gaps in measures of association of TMP/SMX-associated neutropenia in adults living with HIV. We encourage researchers to continue investigating this area to determine if there is a significant effect that could result in clinical benefit.

We have also updated the affiliations of the authors.

Thank you for your support.

Best regards,

Dr. Renata Báez-Saldaña

Hospitalization Service

National Institute of Respiratory Diseases

Tlalpan # 4502, Col. Seccion XVI, C.P. 14080, Mexico, City. Mexico

Phone (52- 55) 5487 17 00 ext. 5284

Email: baezrd@unam.mx

---

## [Decision Letter · Decision Letter 1]

26 Apr 2023

Trimethoprim-Sulfamethoxazole-associated early neutropenia in Mexican adults living with HIV: A cohort study.

PONE-D-22-34622R1

Dear Dr. Báez-Saldaña,

We’re pleased to inform you that your manuscript has been judged scientifically suitable for publication and will be formally accepted for publication once it meets all outstanding technical requirements.

Kind regards,

Dickens Otieno Onyango

Academic Editor

PLOS ONE

Additional Editor Comments (optional):

Reviewers' comments:

Reviewer's Responses to Questions

**Comments to the Author**

1. If the authors have adequately addressed your comments raised in a previous round of review and you feel that this manuscript is now acceptable for publication, you may indicate that here to bypass the “Comments to the Author” section, enter your conflict of interest statement in the “Confidential to Editor” section, and submit your "Accept" recommendation.

Reviewer #1: (No Response)

Reviewer #2: All comments have been addressed

2. Is the manuscript technically sound, and do the data support the conclusions?

Reviewer #1: (No Response)

Reviewer #2: Yes

3. Has the statistical analysis been performed appropriately and rigorously? 

Reviewer #1: (No Response)

Reviewer #2: Yes

4. Have the authors made all data underlying the findings in their manuscript fully available?

Reviewer #1: (No Response)

Reviewer #2: Yes

5. Is the manuscript presented in an intelligible fashion and written in standard English?

Reviewer #1: (No Response)

Reviewer #2: Yes

6. Review Comments to the Author

Reviewer #1: (No Response)

Reviewer #2: Great revisions made to this second version. A few more suggestions for your consideration:

Abstract:

Remove “Aimed to identify” and just say “identified”

Add something about the control group not being denied access to TMP/SMX but voluntarily chose not to take it. You need to be careful that general care guidelines aren’t perceived as being denied to patients.

Make sure you use either “TMP/SMX” or “TMP-SMX” consistently – don’t use both

Introduction:

1st sentence – add “and other opportunistic infections” (?)

2nd sentence interrupts the flow – you want to keep the TMP focus of sentence one flowing into sentence 3 – so remove the 2nd sentence and remove the paragraph break

Methods:

Ensure the voluntary stop/washout period of TMP use is clear – make sure it doesn’t read as though routine care was denied.

Results:

Table formatting needs improvement – see examples in other published papers in this journal

7. PLOS authors have the option to publish the peer review history of their article (what does this mean?). If published, this will include your full peer review and any attached files.

Reviewer #1: No

Reviewer #2: **Yes: **Beth A Tippett Barr

---

## [Editor Report · Acceptance letter]

3 May 2023

PONE-D-22-34622R1 

Trimethoprim-Sulfamethoxazole-associated early neutropenia in Mexican adults living with HIV: A cohort study. 

Dear Dr. Báez-Saldaña:

I'm pleased to inform you that your manuscript has been deemed suitable for publication in PLOS ONE. Congratulations! Your manuscript is now with our production department. 

Kind regards, 

on behalf of

Dr. Dickens Otieno Onyango 

Academic Editor

PLOS ONE